# Maspin is a PTEN-Upregulated and p53-Upregulated Tumor Suppressor Gene and Acts as an HDAC1 Inhibitor in Human Bladder Cancer

**DOI:** 10.3390/cancers12010010

**Published:** 2019-12-18

**Authors:** Yu-Hsiang Lin, Ke-Hung Tsui, Kang-Shuo Chang, Chen-Pang Hou, Tsui-Hsia Feng, Horng-Heng Juang

**Affiliations:** 1Department of Urology, Chang Gung Memorial Hospital-Linkou, Kwei-Shan, Tao-Yuan 33302, Taiwan; laserep@mail.cgu.edu.tw (Y.-H.L.); t2130@cgmh.org.tw (K.-H.T.); glucose1979@gmail.com (C.-P.H.); 2Graduate Institute of Clinical Medical Science, College of Medicine, Chang Gung University, Kwei-Shan, Tao-Yuan 33302, Taiwan; 3Department of Anatomy, College of Medicine, Chang Gung University, Kwei-Shan, Tao-Yuan 33302, Taiwan; D0501301@stmail.cgu.edu.tw; 4Graduate Institute of Biomedical Sciences, College of Medicine, Chang Gung University, Kwei-Shan, Tao-Yuan 33302, Taiwan; 5School of Nursing, College of Medicine, Chang Gung University, Kwei-Shan, Tao-Yuan 33302, Taiwan; thf@mail.cgu.edu.tw

**Keywords:** bladder, maspin, HDAC1, p53, PTEN, tumorigenesis

## Abstract

Maspin is a member of the clade B serine protease inhibitor superfamily and exhibits diverse regulatory effects in various types of solid tumors. We compared the expressions of maspin and determined its potential biological functions and regulatory mechanisms in bladder carcinoma cells in vitro and in vivo. The results of RT-qPCR indicated that maspin expressed significantly lower levels in the bladder cancer tissues than in the paired normal tissues. The immunohistochemical assays of human bladder tissue arrays revealed similar results. Maspin-knockdown enhanced cell invasion whereas the overexpression of maspin resulted in the opposite process taking place. Knockdown of maspin also enhanced tumorigenesis in vivo and downregulated protein levels of acetyl-histone H3. Moreover, in bladder carcinoma cells, maspin modulated HDAC1 target genes, including cyclin D1, p21, MMP9, and vimentin. Treatment with MK2206, which is an Akt inhibitor, upregulated maspin expression, whereas PTEN-knockdown or PTEN activity inhibitor (VO-OHpic) treatments demonstrated reverse results. The ectopic overexpression of p53 or camptothecin treatment induced maspin expression. Our study indicated that maspin is a PTEN-upregulated and p53-upregulated gene that blocks cell growth in vitro and in vivo, and may act as an HDAC1 inhibitor in bladder carcinoma cells. We consider that maspin is a potential tumor suppressor gene in bladder cancer.

## 1. Introduction

A recent epidemiologic study indicated that bladder cancer is the 9th most common cancer worldwide [1]. A number of risk factors for bladder cancer development are well known. However, the recurrence and mortality rates associated with bladder cancer remain high, which pertains to a lack of effective strategies for its early detection [2]. It has claimed that the recurrence rates of bladder cancer are related primarily to its biologic nature and this cancer is short of effective intravesical therapies. Developing new biomarkers for bladder cancer detection and understanding the molecular mechanisms of new repressor genes are critical not only for diagnosing purposes but also for selecting new agents as well as predicting therapeutic effectiveness.

Maspin belongs to the serine protease inhibitor superfamily and has diverse effects on different types of cancers [3,4]. Maspin was first discovered as a gene downregulated in invasive and metastatic breast cancer [5]. Since then, reports have indicated different expressions of maspin in cancers of several other visceral origins, including prostate, ovary, lung, thyroid, and colon cancers [6]. However, controversial studies concerning the expression and function of maspin in bladder cancer have been conducted. Some studies have indicated that maspin expression can play a possible clinical role as a novel tumor suppressor gene in the bladder’s transitional cell carcinoma and can be a useful prognostic marker for predicting the tumor behavior in stage T1 bladder tumors [7,8]. Strong expression of maspin was observed in normal urothelium, and the expression of maspin significantly decreased in invasive carcinoma. It concluded that maspin re-expression may become a therapeutic option in treating bladder cancer [9]. However, other early studies have indicated that maspin does not appear to be a promising prognostic marker for bladder cancer. Moreover, maspin may contribute to bladder cancer development because its expression is positively correlated with the tumor grade [10,11]. These contrary results are confirmed not only by in vivo immunohistochemical (IHC) assays but also by in vitro cell models from different laboratories. The report has indicated that expression of maspin is extremely low in T24 cells and is induced by a DNA methylation inhibitor known as 5-aza-2’-deoxycytidine [12]. Furthermore, moderate surface-bound maspin has been observed in bladder papilla RT-4 cells [13]. The two reports showed that p53 wild-type lower grade transitional carcinoma cells expressed higher maspin levels than p53-null poorly differentiated transitional bladder carcinoma cells in vitro. Additionally, one study revealed that RT-4 cells are maspin-negative [11]. 

A study demonstrated that histone deacetylase 1 (HDAC1), which is a major HDAC responsible for histone deacetylation, interacted with maspin in a yeast two-hybrid screening, and the endogenously expressed maspin and purified maspin inhibited HDAC1 activity [14]. Another study indicated a unique inhibitory interaction of maspin, but not other serpins, with HDAC1 [3]. Aspartate (346) is a critical cis-element in the maspin sequence that determines the molecular context and subcellular localization of maspin, which correlates with HDAC1 expression in human prostate and breast cancer cells [15,16,17]. Therefore, maspin acts as an antitumor gene by modifying the downstream genes of HDAC1. 

The objectives of this study were to determine the expression, function, and regulation of maspin in bladder cancer. We compared the expressions of maspin in human bladder tissues and determined its potential biologic functions and regulatory mechanisms in bladder carcinoma cells. 

## 2. Results

### 2.1. Identification of Maspin as an Antitumor Gene for Human Bladder Cancer 

To verify the role of maspin on tumor suppression, we compared mRNA and protein expressions of maspin in human bladder tissues and bladder carcinoma cells. The RT-qPCR analysis of paired human bladder tissues revealed that the mean of ΔΔC_t_ was 1.9812 ± 0.7575 (when using β-actin as an internal control, Figure 1A) and 2.0258 ± 0.7701 (when using 18S as internal control, Figure 1B) between normal and cancerous tissues. This suggests that normal bladder tissues have significantly higher maspin mRNA levels than bladder cancerous tissues. Although the present evidence is not enough to correlate maspin mRNA expression to the clinic-pathologic characteristics of the patients due to a small sample size, there was further IHC staining (Figure 1C, Appendix A) for maspin in human bladder tissue arrays with normal and cancerous tissues (grade I–III). Qualitative intensity scores of maspin immunostaining in tissues displayed that normal bladder tissues expressed significantly higher maspin protein levels than bladder cancerous tissues (Figure 1D). Further studies engaging a larger sample size is warranted.

### 2.2. Expressions of Maspin in Bladder Carcinoma Cells 

Results of immunoblot (Figure 1E) and RT-qPCR (Figure 1F) assays indicated that TSGH-8301 cells expressed the highest maspin protein (Figure 1E) and mRNA levels (Figure 1F) than the other three carcinoma cell lines, which include RT-4, HT1376, and T24. 

### 2.3. Effects of Maspin on Cell Proliferation in Bladder Carcinoma Cells

We evaluated the modulation of maspin on cell proliferation. The immunoblot assays confirmed the maspin-knockdown in RT-4 cells (Figure 2A) and ectopic overexpression of maspin in T24 cells (Figure 2B). The results of EdU flow cytometry (Figure 2C) indicated that the knockdown of maspin (RT4_shMaspin) increased the amount of cells with EdU incorporation by 15% compared with mock-transduced (RT4_shCOL) cells. The opposite results were observed in the ectopic overexpression of maspin in T24 cells by decreasing 14% of EdU incorporation (Figure 2D). EdU staining proliferation assays also yielded similar results to those of EdU flow cytometry (Appendix A). A qualitative analysis indicated that knockdown maspin in RT-4 cells enhanced the amount of cells with EdU staining by 27% (Appendix A). As determined in EdU staining proliferation assays (Appendix A), when T24 cells exhibited an overexpression of maspin, the percentage of positive EdU staining cells decreased by 22% when compared with mock-transfected T24 (T24-DNA) cells. 

### 2.4. Effects of Maspin on Cisplatin-Induced Apoptosis in Bladder Carcinoma T24 Cells

We continued to evaluate the effect of maspin on cell apoptosis. Further MTS cell viability assays indicated that cisplatin treatment decreased cell viability in a dose-dependent manner in T24-DNA cells. Overexpression of maspin considerably reduced resistance to cisplatin treatment (Figure 2E). Flow cytometric analysis with double fluorescence staining (Annexin V-FITC/PI) revealed a significantly higher number of apoptotic cells in T24-maspin cells than in T24-DNA cells (18.41 ± 1.26 vs. 5.03 ± 0.85) after 24 h of 40-µM cisplatin treatment (Figure 2F). 

### 2.5. Effects of Maspin on Cell Invasion in Bladder Carcinoma Cells

To determine the effect of maspin on cell invasion, we knocked down maspin in HT1376 cells and ectopic-overexpressed maspin in T24 cells. Through immunoblot assays, we confirmed that the expression of maspin was approximately 15% in maspin-knockdown HT1376 (HT_shMaspin) cells when compared with mock-knockdown (HT_shCOL) cells (Figure 3A, top). The results of Matrigel invasion assays indicated that knockdown of maspin resulted in a 1.5-fold increase in the invasion capacity when compared with HT_shCOL cells (Figure 3A, bottom). The immunoblot assays demonstrated maspin ectopic overexpression in T24 (T24-maspin) cells when compared with mock-transfected T24 (T24-DNA) cells (Figure 3B, top). By contrast, the invasion capacity was downregulated by 65% in T24-maspin cells when compared with T24-DNA cells (Figure 3B, bottom).

### 2.6. Effect of Maspin-Knockdown on the Tumorigenesis of Bladder Carcinoma HT1376 Cells

The effect of maspin on tumor growth in vivo was evaluated using xenografts of BALB/cAnN-Foxn1^NU^ mice. The tumors generated from mock-transducted HT1376 (HT-shCOL) cells (Figure 4A) grew slower than those derived from maspin-knockdown HT1376 (HT_shMaspin) cells (Figure 4B). Moreover, the tumors generated from HT_shMaspin cells were approximately 2.68-fold larger than the tumors generated from HT_shCOL (178.91 ± 41.39 mm^3^ vs. 66.58 ± 16.31 mm^3^) after 36 days of growth (Figure 4C). The weight of tumors derived from HT_shMaspin cells was approximately 1.7 times higher than the weight of tumors derived from HT_shCOL cells (0.26 ± 0.02 vs. 0.15 ± 0.03 g, Figure 4D). We randomly selected three tissues from each group to perform immunoblot assays, and the results confirmed that the expression of maspin was significantly lower in the xenograft tumors derived from HT_shMaspin cells than in those derived from HT_shCOL cells (Figure 4E). Furthermore, RT-qPCR assays revealed that the expressions of maspin and p21 were significantly lower in tumors derived from HT_shMaspin cells than in tumors derived from HT_shCOL cells. By contrast, the expressions of cyclin D1, vimentin, MMP2, and MMP9 were higher in tumors derived from HT_shMaspin cells than in tumors from HT_shCOL cells (Figure 4F).

### 2.7. Maspin as an HDAC1 Inhibitor in Bladder Carcinoma Cells

To understand whether maspin serves as an endogenous inhibitor of HDAC1 activity in bladder carcinoma cells, we determined the modulation of maspin on HDAC1 activity and evaluated the downstream genes of maspin in RT-4, HT1376, or T24 cells. The immunoblot assays indicated that the expression of acetyl-H3 was blocked in RT4_shMaspin cells and HT_shMaspin cells but enhanced in T24-maspin cells when compared with mock-treated wild-type cells. By contrast, the expression of HDAC1 was reversed in the previously mentioned cells (Figure 5A). The results of HDAC1 activity assays indicated that knockdown of maspin induced HDAC1 activity in RT-4 cells (Figure 5B), whereas the overexpression of maspin clearly blocked HDAC1 activity in T24 cells (Figure 5C). The net HDAC1 activity of RT4_shCOL, RT4_shMaspin, T24-DNA, and T24-Maspin cells were 230.2 ± 3.5, 302.5 ± 4.0, 324.1 ± 4.1, and 217.8 ± 3.4 pmol/min/mg protein, respectively (Appendix A). Further immunoblot assays revealed that expressions of maspin and p21 were significantly lower in RT4_shMaspin cells and HT_shMaspin cells but higher in T24-maspin cells when compared with their mock-treated wild-type cells. By contrast, the expressions of cyclin D1, MMP9, and vimentin were reversed in the previously mentioned cells (Figure 5D). RT-qPCR assays also revealed that the expressions of maspin and p21 were significantly lower in HT_shMaspin cells than in HT_shCOL cells. However, the expressions of cyclin D1, vimentin, and MMP9 were enhanced when maspin was knocked down in HT1376 cells (Figure 5E).

### 2.8. Effect of PTEN on the Tumorigenesis and Maspin Expression in Bladder Carcinoma Cells

We evaluated whether maspin is the downstream gene of PTEN in bladder carcinoma cells. The immunoblot assays indicated that knockdown of PTEN in RT-4 cells downregulated maspin expression, whereas ectopic overexpression of PTEN in T24 cells reversed this effect (Figure 6A). The [^3^H]thymidine incorporation assays revealed that PTEN-knockdown enhanced cell proliferation in RT-4 cells (Figure 6B). Further analysis using xenografts of BALB/cAnN-Foxn1^NU^ mice revealed that tumors generated from PTEN-knockdown RT-4 (RT4_shPTEN) cells grew faster than those derived from mock-transduced RT-4 (RT4_shCOL) cells (Figure 6C). Tumors generated from RT4_shPTEN cells were approximately six-fold larger than those generated from RT4_shCOL cells (58.53 ± 12.34 mm^3^ vs. 10.75 ± 0.67 mm^3^) after seven weeks of growth (Figure 6D). Since the most known function of PTEN is the negative regulator of the Akt (protein kinase B) pathway, we continued to determine the effect of PTEN on Akt phosphorylation and maspin expression. Additional immunoblot assays demonstrated that PTEN-knockdown in RT-4 cells upregulated Akt phosphorylation at both S473 and T308 (Figure 6E). Moreover, we treated RT-4 cells with VO-OHpic trihydrate, which is a type of PTEN activity inhibitor. After treatment, the expression of p-Akt (T308 and S473) increased, the expression of maspin decreased, and the expressions of PTEN and Akt remained the same (Figure 6F). The RT-qPCR assays for RT-4 (Figure 6G) and T24 (Figure 6H) cells indicated that knockdown of PTEN in RT-4 cells downregulated maspin expression, whereas ectopic overexpression of PTEN in T24 cells reversed this effect. We also treated T24 cells with MK2206, which is an allosteric Akt inhibitor. After treatment, the expression of p-Akt (T308 and S473) decreased, the expression of maspin increased, and the expression of Akt remained the same (Figure 6I). The reporter assay for the maspin reporter’s vector-cotransfected HT1376 cells with varied concentrations of the PTEN expression vector revealed that maspin reporter activity was induced by PTEN (Figure 6J). In summary, our results indicated that maspin expression in human bladder cancerous cells was stimulated by PTEN.

### 2.9. p53 Upregulates Maspin Expression in Bladder Carcinoma Cells

The immunoblot (Figure 7A) and RT-qPCR (Figure 7B) assays indicated overexpression of p53 induced maspin expression in p53-mutant HT1376 cells. Camptothecin (Cpt) treatment (1 µM) induced p53 and maspin expressions, whereas p53-knockdown attenuated these effects in p53 wild-type RT-4 cells (Figure 7C). The results from the reporter assays indicated that the cotransfected p53 expression vector induced the reporter activity of the human maspin gene in HT1376 cells (Figure 7D). Moreover, 5′-deletion report assays revealed that p53-induced maspin reporter activity was dependent on the DNA fragments (−573 to −363 and −363 to −162) of the 5′-flanking region of the human maspin gene (Figure 7E). The transient cotransfected p53 expression vector did not affect the reporter’s activity of a reporter vector containing the SV40 promoter.

## 3. Discussion

Maspin belongs to the serine protease inhibitor/noninhibitor superfamily and has dissimilar effects according to the types of cancer [4]. Although reports concerning the biological function of maspin in bladder cancer are still contradictory, our previous studies have indicated that maspin is the downstream gene of the prostate-derived Ets factor (PDEF) and growth differentiation factor 15 (GDF15), which are antitumor genes in bladder carcinoma cells [18,19]. 

Our analysis of paired human bladder tissues revealed that normal bladder tissues express significantly higher maspin mRNA levels than bladder cancerous tissues. IHC staining in human bladder tissue arrays also revealed that normal bladder tissues expressed significantly higher maspin protein levels than bladder cancerous tissues (Figure 1). These results indicated that *maspin* can be regarded as an antitumor gene in bladder cancer, which is in agreement with previous studies from several independent laboratories [7,8,9,20,21]. However, our in vitro study results revealed that TSGH-8301 cells expressed the highest maspin protein levels among the four carcinoma cell lines (RT-4, HT1376, T24, and TSGH-8301), which indicated that maspin expression among the bladder carcinoma cell lines is dependent on the cell type but is not related to the extent of neoplasia in vitro. This result is similar to a study of prostate carcinoma cells, which also discovered that maspin expression is dependent on the cell type and is not related to the extent of neoplasia in vitro [22]. A previous study indicated that the function of maspin is associated with its subcellular locations [15]. Therefore, further investigation in precise mechanisms is warranted regarding whether the various subcellular distributions of maspin influence HDAC1 activity in bladder carcinoma cells.

The results of this study demonstrated that maspin attenuated cell proliferation and invasion in bladder carcinoma cells in vitro (Figure 2 and Figure 3), which agrees with another study [13]. Our results clearly demonstrated that knockdown maspin in p53 wild-type lower grade transitional carcinoma RT-4 cells grew faster than mock-knockdown RT-4 cells. At the same time, ectopic overexpression of maspin in p53-null poorly differentiated transitional bladder carcinoma T24 cells attenuated cell growth in vitro. Additional MTS and apoptosis assays revealed that ectopic maspin overexpressed in T24 (T24-Maspin) cells was more sensitive to cisplatin-induced apoptosis than mock-transfected T24 cells were (Figure 2E,F). These results are consistent with those of other studies reporting that the ectopic overexpression of maspin improved the sensitivity of bladder carcinoma cells to cisplatin [23,24]. Moreover, our study demonstrated that maspin blocked tumor growth in vivo when using xenografts of BALB/cAnN-Foxn1^NU^ mice (Figure 4). RT-qPCR assays revealed that the expression of maspin affected the gene expressions of cyclin D1, p21, vimentin MMP2, and MMP9 in xenograft tumors, which suggested that maspin may modulate these genes to regulate tumor growth in vivo. A previous study indicated that maspin blocked MMP9 expression in prostate carcinoma cells [25]. The previously mentioned genes are also potential target genes of HDAC1 indicating that maspin may correlate with the HDAC1 activity.

Reports have indicated that maspin serves as an endogenous inhibitor of HDAC1 activity, which leads to the induction of apoptosis in several cancer cells [14,26,27,28,29,30]. Although pathological examination revealed that HDAC1 was significantly associated with high bladder cancer tumor grades [31], whether maspin acts as an HDAC1 inhibitor in bladder carcinoma cells remains unclear. Our results indicated that knockdown maspin induced a small but significant expression of HDAC1 while downregulating the protein levels of Aetyl-H3, which suggests that maspin acts as the inhibitor of HDAC1. HDAC1 activity assays also clearly indicated that maspin blocked the net HDAC1 activity in bladder carcinoma cells (Figure 5). These results align with previous studies in the prostate carcinoma cells [14,22]. The results of immunoblot and RT-qPCR assays indicated that maspin upregulated p21 but downregulated cyclin D1, MMP9, and vimentin gene expressions in vitro. The p21, p27, cyclin D1, MMP2, MMP9, and vimentin genes have been identified as HDAC1 downstream genes [32,33,34,35,36]. The similar in vivo results are illustrated in Figure 4. Our results are the first to conclude that maspin may act as an HDAC1 inhibitor in bladder carcinoma cells in vitro and in vivo. However, it is necessary to further investigate the precise mechanisms involved in the maspin/HDAC1 signaling axis in bladder cancer.

Phosphatase and a tensin homolog deleted on chromosome 10 (PTEN) has been widely known as a tumor suppressor gene, and the loss of PTEN expression is correlated with disease invasiveness in bladder cancer [37]. A previous study reported that the ectopic overexpression of PTEN induced maspin expression in human glioblastoma U87 cells [38]. However, no report has been published on the effect of PTEN on maspin expression in bladder cancer. The results of the present study indicated that PTEN modulated the cell growth of bladder carcinoma cells in vitro and in vivo, which is consistent with the results of other studies [39,40]. The immunoblot assays revealed that PTEN downregulated Akt phosphorylation at both S473 and T308, whereas treatment with VO-OHpic trihydrate, which is a PTEN activity inhibitor, reversed the results. The treatment of T24 cells with MK2206, which is an Akt activity inhibitor, upregulated maspin gene expression (Figure 6). Thus, our results clarified that PTEN enhanced maspin expression through the Akt activity. In summary, our results indicated that maspin expression in human bladder carcinoma cells was stimulated by PTEN. 

A previous report indicated that maspin is a p53-upregulated gene in prostate carcinoma cells [41]. Our study using the ectopic overexpression of p53 in p53-mutant HT1376 cells and camptothecin treatment induced p53 in p53 wild-type RT-4 cells demonstrated that maspin was upregulated by p53 in bladder carcinoma cells. The 5′-deletion report assays demonstrated that p53-induced maspin expression in bladder carcinoma cells was dependent on the 5′-flanking region fragment (−567 to −162) of the human *maspin* gene (Figure 7). Our study is consistent with a previous report that identified two p53 binding sites (GGCATGTTGGAGGCCTTTG and GGACAAGCTGCCAAGAGGCTTGAGT) in the promoter of the human maspin gene [40]. However, our study indicated that p53 may affect the reporter activity of the reporter vector containing the DNA fragment (−567 to −363), which does not have a putative p53 binding site. The majority of gene expressions modulated by p53 generally occurred through consensus sequences other than that of the p53 DNA-binding site [42,43]. Our results confirmed that maspin is a target gene of PTEN and p53, and should be referred to as an anti-tumor gene in bladder carcinoma cells. 

## 4. Materials and Methods

### 4.1. Materials, Cell Lines, and Cell Culture

The bladder transitional cell carcinoma cell lines including the RT-4, HT1376, TSGH-8301, and T24 cell lines were purchased from the Bioresource Collection and Research Center (Hsinchu, Taiwan) and cultured as described in a previous study [44]. The RT-4 cell line was derived from explants of a recurrent papillary bladder tumor. The HT1376 cell line was derived from a Caucasian woman with grade 3 transitional cell bladder cancer. The TSGH-8301 cell line was derived from the well-differentiated transitional cell carcinoma of a Taiwanese patient. The T24 cell line was composed of poorly differentiated transitional carcinoma cells with low tumorigenic capability. The RT-4 cells expressed p53-wild-type and PTEN-wild-type genes. The HT1376 and T24 cells expressed p53-mutant and PTEN-mutant type genes. The 4,6-Diamino-2-phenylindole (DAPI) and bovine serum albumin (BSA) were obtained from Sigma-Aldrich Co. (St. Louis, MO, USA). Fetal calf serum (FCS) was purchased from HyClone (Logan, UT, USA). RPMI 1640 media were obtained from Invitrogen (Carlsbad, CA, USA). Matrigel was purchased from BD Biosciences (Bedford, MA, USA). MK2206, VO-OHpic trihydrate, cisplatin, and camptothecin (Cpt) were purchased from Sigma-Aldrich Co (St. Louis, MO, USA). 

### 4.2. Tissue Collection and Analysis

The specimens for human paired bladder tissue biopsies were obtained from patients admitted to the Department of Urology, Chang Gung Memorial Hospital-Linkou (Tao-Yuan, Taiwan). The bladder tumor samples were derived from endoscopic resections and the normal bladder tissues were collected from the same patients with bladder cancer. The bladder tissues were classified according to pathological examinations of the parallel preparations from respective samples by attending pathologists. The Institutional Review Board of the Chang Gung Memorial Hospital approved the protocol for tissue collection and analysis (Approval: IRB 103-33504B).

### 4.3. Immunohistochemical Assays

The human bladder tissue array was purchased from Abcam (Cat no: ab178086, Cambridge, MA, USA). The pathological staging in the grading of the tumors was based on the data sheet provided by the manufacturer. The tissue sections were stained using the Bond-Max autostainer (Leica Biosystems, Singapore). Slides were dewaxed in Bond Dewax solution and hydrated in Bond Wash solution. Antigen retrieval was performed at an acidic pH by using Epitope Retrieval 1 solution for 30 min at 100 °C. The slides were then incubated with the primary antibody (anti-maspin, cat# 554292, BD Bioscience, Bedford, MA, USA) at a concentration of 1:100 for 30 min at room temperature. The detection kit used was the Bond Polymer Refine Detection (DS9800, Leica Biosystems). The incubations with the post-primary antibody, the polymer, DAB, and hematoxylin lasted 8, 8, and 5 min, respectively. Following staining on the instrument, the slides were dehydrated using graded alcohols to xylene and cover-slipped with the mounting medium. Images were captured using a Paxcam 3 camera (PAX-it, Villa Park, IL, USA). The intensity score (0 = none, 1 = weak, 2 = medium, 3 = strong, 4 = intense) of the images was then analyzed. 

### 4.4. Gene Knockdown

Cells were plated onto six-well plates one day before analysis. The culture media were replaced with RPMI-1640 medium plus 10% FCS and 5 μg/mL polybrene (Santa Cruz Biotechnology, Santa Cruz, CA, USA) and transduced with maspin shRNA Lentiviral particles (sc-35859, Santa Cruz Biotechnology). Two days after transduction, the cells were selected by incubating with 10 μg/mL puromycin dihydrochloride for at least another three generations, as described in a previous study [19]. The p53-knockdown RT-4 (RT4_shp53), PTEN-knockdown RT-4 (RT4_shPTEN) and mock-knockdown RT-4 (RT4_shCOL) cells were cloned as described previously [39].

### 4.5. Expression Vector Constructs and Stable Transfection

The expression vector with full-length human maspin cDNA was purchased from Invitrogen. Electroporation was conducted with the ECM 830 system (BTX, San Diego, CA, USA) by setting the voltage at 180 V, setting the pulse length at 90 milliseconds, and using a single pulse setting, as described in a previous method [18]. The mock-transfection cells were transfected with a control of a pcDNA3 expression vector and clonally selected in the same manner as the gene-overexpressed cells. The p53 and PTEN expression vectors were constructed. p53-overexpressed HT1376 (HT-p53) cells and PTEN-overexpression T24 (T24-PTEN) cells were cloned as described previously [40].

### 4.6. Immunoblot Assay

The blotting membranes were probed with antiserum of maspin (554292, BD Bioscience), MMP9 (EP1254, Abcam), vimentin (AJ1815a; Abgent, San Diego, CA, USA), HDAC1 (sc-81598), p53 (DO-1, Santa Cruz Biotechnology), p21 (DCS60), cyclin D1 (DCS6), PTEN (#9552), AKT (#4691), pAKT^Ser473^ (#9271), pAKT^Thr308^ (#2965, Cell Signaling Technology Danvers, MA, USA), acetyl-H3 (06-599), or β-Actin antiserum (MAB1501, Merck Millipore, Burlington, MA, USA). Band intensities were recorded using the Chemi Genius II BioImaging System of Syngene (Cambridge, UK) and analyzed using the GeneTool Program of ChemiGenius (Syngene).

### 4.7. Reverse Transcription Real-Time Polymerase Chain Reaction

The total RNA was isolated with a Trizol reagent, and cDNA was synthesized using the superscript III preamplification system (Invitrogen). The real-time polymerase chain reaction (PCR) was performed using a CFX Connect Real-Time PCR system (Bio-Rad Laboratories, Foster city, CA, USA). The FAM dye-labeled TaqMan MGB probes as well as PCR primers for human maspin (Hs00985283_m1), p21 (Hs00355782_m1), p27 (Hs01597588-m1), MMP2 (Hs01548728_m1), MMP9 (Hs00234579_m1), vimentin (Hs00185584_m1), Cyclin D1 (Hs00277039_m1), PTEN (Hs02621230_sl), p53 (Hs01034249_m1), 18S (Hs03003631_g1), and β-actin (Hs01060665_g1) were purchased from Applied Biosystems (Foster City, CA, USA). Either 18S or β-actin was used as an internal positive control. The mean cycle threshold (C_t_) values for target genes were normalized against the 18S or β-actin control probe to calculate the ΔC_t_ values by using the StepOne software program v2.0 (Applied Biosystems). All the reactions were conducted in triplicate, and each experiment was conducted on at least three independent occasions. 

### 4.8. HDAC1 Activity Assay

The HDAC1 activity was measured in nuclear extracts prepared from bladder carcinoma cells by using the HDAC1 Immunoprecipitation and Activity Assay Kit (BioVision, Milpitas, CA, USA). The nuclear extracts of the cells were obtained as per a previous method [45]. Subsequently, 30 µg of protein was immunoprecipitated with HDAC1 antibody or rabbit IgG as a control antibody. After bead capture, the samples were incubated with the HDAC1 assay buffer and substrates. The total HDAC1 activity was measured fluorometrically. Fluorescence was read at 380/500 nm, and the results were illustrated using a standard curve, according to the manufacturer’s instructions and expressed as the pmol/min/mg protein. The background HDAC1 activity of each prepared sample was measured using the same amount of protein immunoprecipitated with rabbit IgG. The net HDAC1 activity of the cells was determined using the following formula: HDAC1 activity = total HDAC1 activity—background. 

### 4.9. Thymidine Incorporation Assays

Cell proliferation was measured using the ^3^H-thymidine incorporation assay, as described in a previous study [46]. Cells were cultured in each well of a six-well plate with 10% FCS. After the required incubation time, 1 µCi/mL of ^3^H-thymidine was added and then incubation was continued for another 4 h. Cells were washed twice in the cold phosphate buffer saline (PBS) and then with the cold 5% trichloroacetic acid. Cells were lysed by adding 0.5 mL of 0.5 N NaOH and then the solubilized cell solution was mixed with a scintillation cocktail and counted in a liquid scintillation analyzer (Packard BioScience, IL, USA). 

### 4.10. Cell Viability

Cell viability was measured with an MTS assay following the manufacturer’s protocol as previously described [46]. Cells (T24-DNA and T24-Maspin) were grown in 100 µL of RPMI 1640 medium with 10% FCS for one day, and then treated with various dosages of cisplatin for another 24 h. Cells were then incubated with freshly prepared MTS/phenzaine methosulfate solution (Progmeg Bioscienesc, San Luis Obispo, CA, USA) for 3 h, and the plate was read at an absorbance of 490 nm using a Synergy H1 microplate reader (BioTek, Beijing, China).

### 4.11. EdU Flow Cytometry Assay

Cells (5 × 10^5^) were cultured in serum-free medium for 24 h. After being incubated for another 48 h in a medium with 10% serum, the cells were incubated with EdU (5-ethynyl-2′-deoxyuridine, 10 µM) for 2 h. Subsequently, the cells were collected and analyzed using Click-iT EdU Flow Cytometry Assay Kits (Thermo Fisher Scientific Inc. Waltham, MA, USA), as described by the manufacturer. The EdU fluorescence of the cells was detected using an Attune NxT acoustic focusing cytometer (Thermo Fisher Scientific Inc.), as described previously [47]. 

### 4.12. EdU Imaging Assay

Cells were seeded on sterile glass coverslips for 24 h and cultured in serum-free medium for 24 h. After incubation for another 48 h in a medium with 10% serum, the cells were treated with EdU (5-ethynyl-2′-deoxyuridine; 50 µM) for 2 h. The cells were fixed in 3.7% paraformaldehyde in phosphate-buffered saline (PBS) at room temperature for 15 min and then washed twice with 3% BSA in PBS. The cells were permeabilized through saponin-based permeabilization and the wash reagent was used for 20 min. Then, the cells were treated with the Click-iT reaction cocktail (Thermo Fisher Scientific Inc.) for 30 min and washed again with 3% BSA in PBS. The coverslips were mounted with a Prolong Gold antifade reagent and DAPI (Thermo Fisher Scientific Inc.). The EdU fluorescence of the cells was recorded and photographed under a microscope (BX43, OLYMPUS, Tokyo, Japan), as described previously [47]. 

### 4.13. Annexin V-FITC Apoptosis Detection

The cell pellets of T24-DNA and T24-maspin cells were harvested. Then, apoptosis detection and quantification were performed after treating the pellets with Annexin V-FITC and propidium iodide (PI) (BioVision Inc., Milpitas, CA, USA) by using the Attune NxT acoustic focusing cytometer (Thermo Fisher Scientific Inc.), as described in a previous study [48]. 

### 4.14. Matrigel Invasion Assay

The invasion ability of the cells was determined through an in vitro Matrigel invasion assay, as described in Reference [19]. Cells that migrated to the other side of the transmembrane were fixed with 4% paraformaldehyde and then stained with 0.1% crystal violet solution for 10 min. The quantity of cells that invaded the Matrigel was recorded microscopically (IX71, Olympus, Tokyo, Japan).

### 4.15. Xenograft Animal Model

Animal studies were conducted in accordance with the United States National Institutes of the Health Guide for the Care and Use of Laboratory Animals and were approved by the Chang Gung University Animal Research Committee. In all procedures, every effort was made to minimize the suffering of the laboratory animals and the number of animals used. Male nude mice (BALB/cAnN-Foxn1) aged four weeks were obtained from the animal center of the National Science Council in Taiwan. Cells were detached from the cell flask through treatment with Gibco Versene solution (Life Technologies, Grand Island, NY, USA) and washed with RPMI1640 medium containing 10% FCS. The cells were then resuspended in a PBS solution. The mice were anesthetized intraperitoneally. In addition, 5 × 10^6^ cell/100 µL cells were injected subcutaneously on the right or left lateral back wall in close proximity to the shoulder of each mouse. The growth of the xenografts was measured using Vernier caliper measurements, which were performed every 2–3 days. The tumor volume was determined using the following formula: volume = π/6 × larger diameter × (smaller diameter)^2^ [19]. 

### 4.16. Reporter Vectors and Reporter Assay

The human maspin reporter vector was constructed by cloning the 5′-DNA fragment (−5948 to −5) of the human maspin gene into the pbGL3 reporter vector (Progmega Biosciences), as previously described [19]. The 5′-deletion nest report vectors were cloned as indicated. The reporter vector containing the Simian virus 40 (SV40) promoter in ppGL3 (pGL3-promoter vector) was purchased from Progmega Biosciences. Cells were seeded at a density of 10^4^ cells/well in a 24-well plate and allowed to grow for 1 day prior to transfection. Cells were transiently cotransfected with reporter vectors and p53 expression vectors as indicated, using the X-tremeGene HP DNA transfection regent (Roche Diagnostics GmbH, Amaaheim, Germany). The luciferase activity was determined using a relative light unit via a Synergy H1 microplate reader (BioTek, Beijing, China) and was adjusted according to the β-galactosidase enzymatic activity previously described [47]. 

### 4.17. Statistical Analysis

The results were expressed as the mean ± standard error (SE) of at least 3 independent replications of each experiment. The statistical significance was determined through paired *t*-test analysis and ANOVA by using the SigmaStat program for Windows version 2.03 (SPSS Inc., Chicago, IL, USA). Multiple comparisons were conducted using ANOVA with Tukey’s post hoc test.

## 5. Conclusions

Our results indicated that the expression of maspin is higher in normal bladder tissues and knockdown maspin enhanced cell growth in bladder carcinoma cells in vitro and in vivo. Our experiments proved that maspin is a PTEN-upregulated and p53-upregulated gene and provided evidence suggesting that maspin is a potential tumor suppressor gene in the human bladder. Maspin acts as an HDAC1 inhibitor, which may modulate the p21, cyclin D1, MMP9, and vimentin expressions in bladder carcinoma cells. The maspin/HDAC1 signaling axis may represent the antitumor characteristics in human bladder carcinoma cells. However, the precise mechanisms involved in the regulation of downstream targets warrant further investigation.

## Figures and Tables

**Figure 1 cancers-12-00010-f001:**
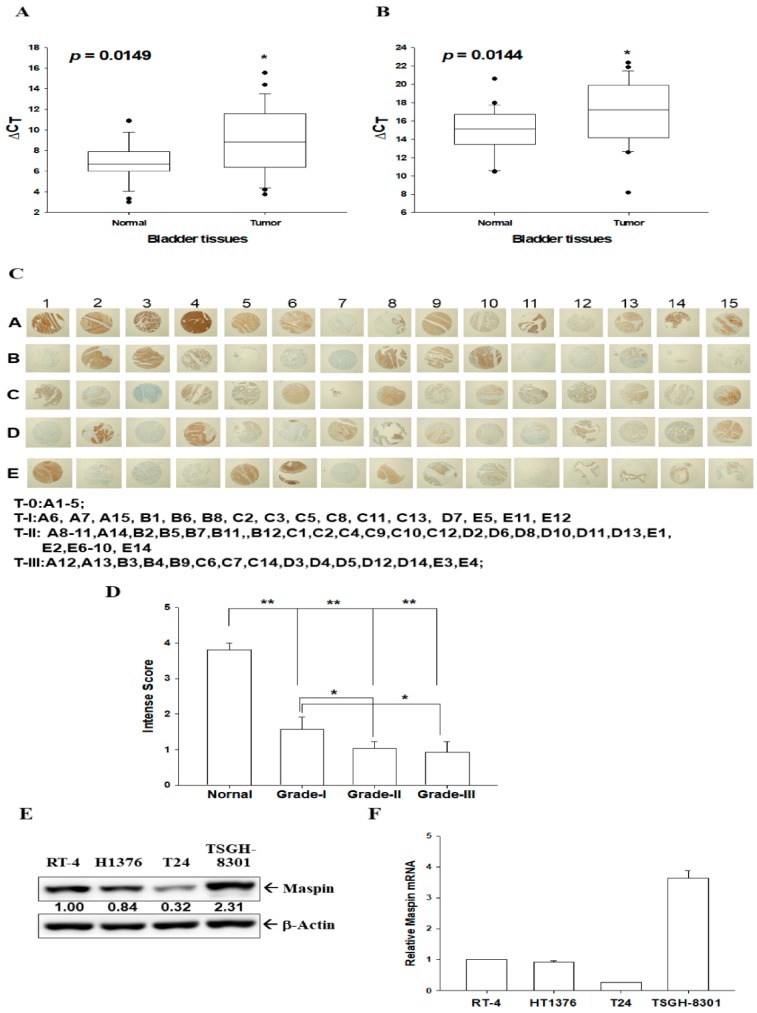
Expression of maspin in human bladder tissues and bladder carcinoma cells. Quantitative analysis of maspin expression in paired bladder cancerous and normal tissues was conducted through RT-qPCR assays by using β-actin (**A**) or 18S (**B**) as an internal control. Box plot analysis was used to compare the maspin expressions in cancerous and normal bladder tissues (*n* = 25). (**C**) IHC staining for maspin in a human bladder tissue array with normal and bladder cancer tissues (grade I, II, and III). (**D**) The intensity scores of maspin immunostaining in normal (*n* = 6) and cancerous bladder tissues (grade TI, *n* = 16, grade TII, *n* = 30, grade TIII, *n* = 15). * *p* < 0.05, ** *p* < 0.01. The expression of maspin in bladder carcinoma cells was determined through (**E**) immunoblot and (**F**) RT-qPCR assays (±SE, *n* = 3). The numbers indicate the ratio of Maspin/β-Actin in relation to RT-4 cells.

**Figure 2 cancers-12-00010-f002:**
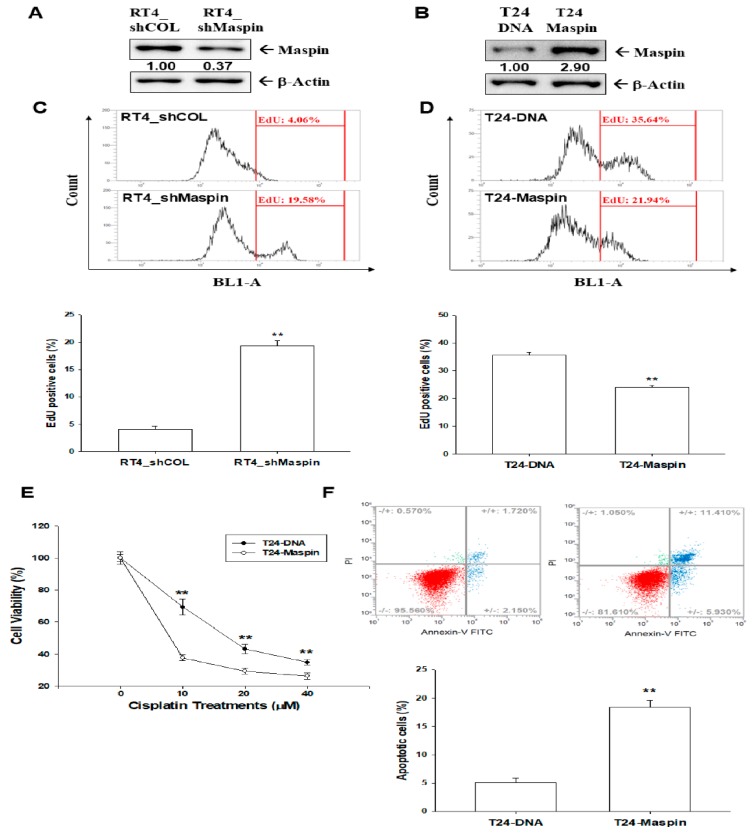
Effects of maspin on cell proliferation and cisplatin-induced apoptosis in bladder carcinoma cells. Protein levels of maspin after knockdown of maspin in RT-4 cells (**A**) and after ectopic maspin overexpression in T24 cells (**B**). The numbers indicate the ratio of maspin/βActin in relation to RT4_shCOL or T24-DNA cells. The proliferation ability of RT4_shCOL, RT4_shMaspin (**C**), T24-DNA, and T24-maspin cells (**D**) was determined through flow cytometry by using the Click-iT EdU flow cytometry kit (±SE, *n* = 4). (**E**) Cell viability of T24-DNA and T24-maspin cells after treatment with various cisplatin levels (±SE, *n* = 8). (**F**) Cells (T24-DNA and T24-maspin) were treated with various concentrations of cisplatin for 24 h. The fluorescence intensity for Annexin V-FITC in conjunction with PI staining was determined through flow cytometry (±SE, *n* = 4). Data are presented as the percentage of apoptotic cells after cisplatin treatment. ** *p* < 0.01.

**Figure 3 cancers-12-00010-f003:**
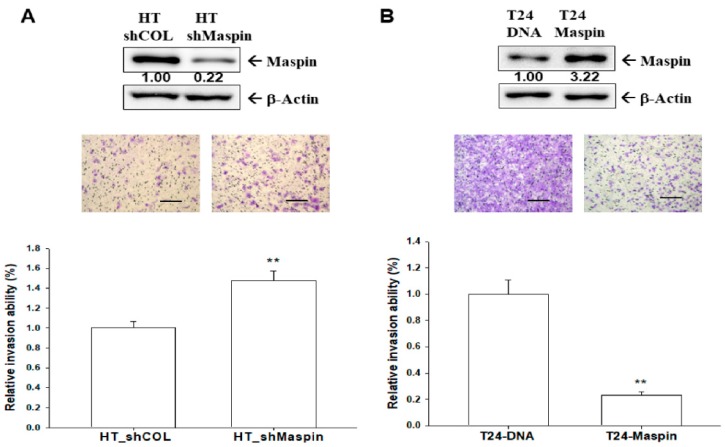
Modulation effect of maspin on cell invasion in bladder carcinoma cells. The invasion ability of cells was determined through in vitro Matrigel invasion assays. Data are presented as a mean percentage (±SE, *n* = 3) in relation to that of the (**A**) HT-shCOL or (**B**) T24-DNA group. The numbers indicate the ratio of Maspin/β-Actin in relation to HT_shCOL or T24-DNA cells. The scale bar is 200 µm. ** *p* < 0.01.

**Figure 4 cancers-12-00010-f004:**
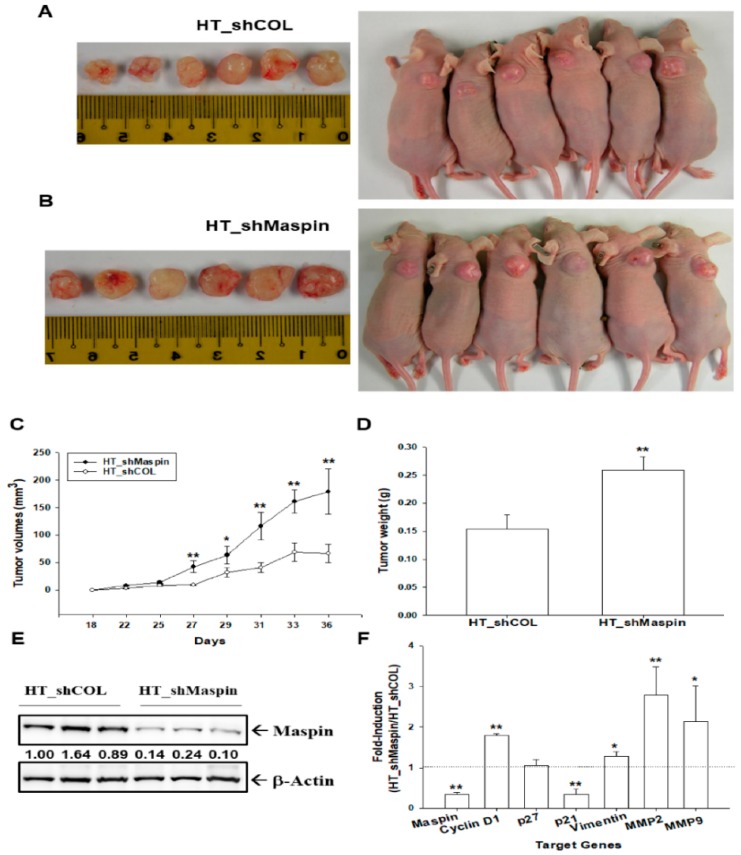
Modulation effect of maspin on tumorigenesis of bladder carcinoma HT1376 cells. Four-week-old male athymic nude mice were divided randomly into two groups. HT_shCOL (**A**) and HT_shMapsin (**B**) cells (5 × 10^6^) were injected subcutaneously in the dorsal area of the mice (*n* = 6). The tumor volumes were measured every two to three days during 36-day periods (**C**). The (**D**) tumor weight (±SE, *n* = 6) and (**E**) protein levels of maspin (*n* = 3) of the tumor derived from HT_shCOL and HT_shMaspin cells were measured after sacrifice. The numbers indicate the ratio of Maspin/β-Actin in relation to the tumor tissue derived from HT_shCO cells. (**F**) The ratio of mRNA levels of maspin, cyclin D1, p21, p27, vimentin, MMP2, and MMP9 between tumors derived from HT_shCOL and HT_shMaspin cells was determined through RT-qPCR assays (±SE, *n* = 6). * *p* < 0.05, ** *p* < 0.01.

**Figure 5 cancers-12-00010-f005:**
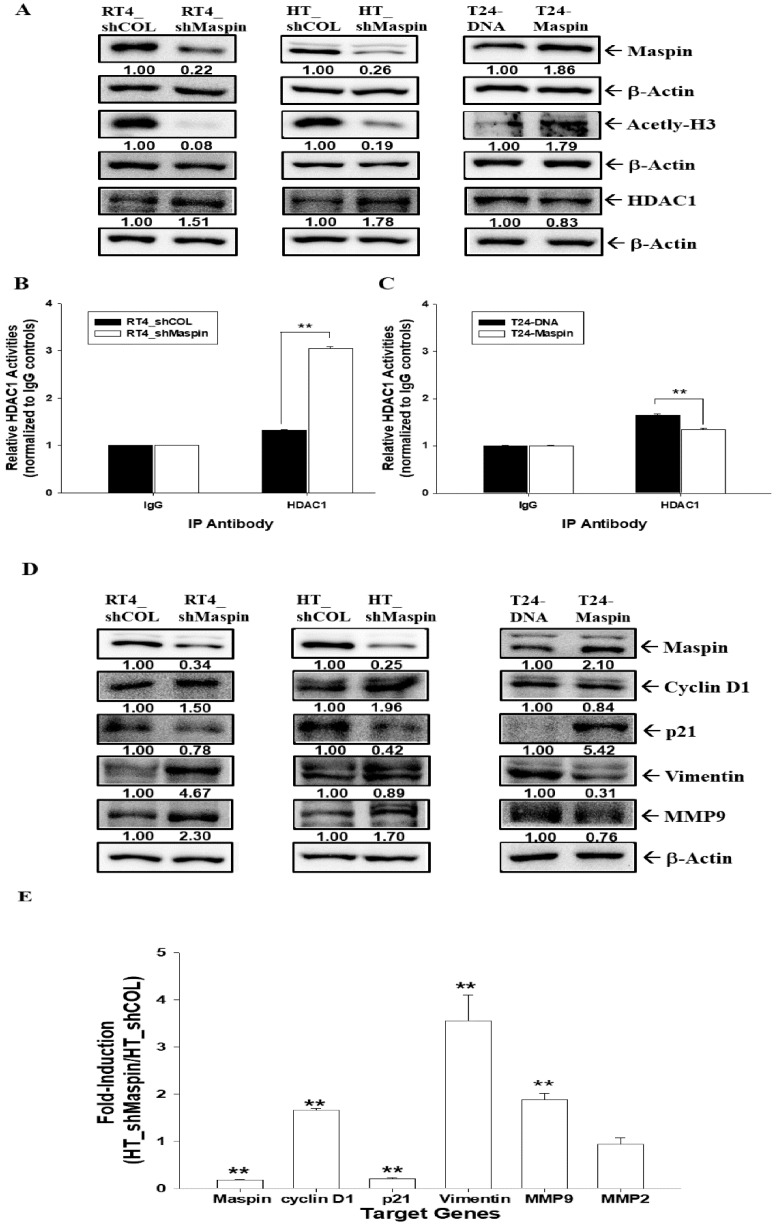
Maspin acts as an HDAC1 inhibitor by modulating downstream genes in bladder carcinoma cells. (**A**) RT4_shCOL, RT4_shMaspin, HT_shCOL, HT_shMaspin, T24-DNA, and T24-maspin cells were lysed, and the protein levels of maspin, acetyl-H3, HDAC1, and β-Actin were determined through immunoblot assays. The RT4_shCOL, RT4_shMaspin (**B**), T24-DNA, and T24-maspin cells (**C**) were extracted, and the net HDAC activity in the nuclei lysis solution was determined by using the HDAC1 immunoprecipitation and activity assay kit (±SE, *n* = 3). (**D**) RT4_shCOL, RT4_shMaspin, HT_shCOL, HT_shMaspin, T24-DNA, and T24-Maspin cells were lysed, and the protein levels of maspin, p21, MMP9, cyclin D1, vimentin, and β-Actin were determined through immunoblot assays. The numbers indicate the ratio of the target gene/β-Actin in relation to RT4_shCOL, HT_shCOL, or T24-DNA cells. (**E**) The mRNA ratio of maspin, cyclin D1, p21, vimentin MMP9, and MMP2 between HT_shCOL and HT_shMaspin cells was determined using RT-qPCR assays (±SE, *n* = 3). ** *p* < 0.01.

**Figure 6 cancers-12-00010-f006:**
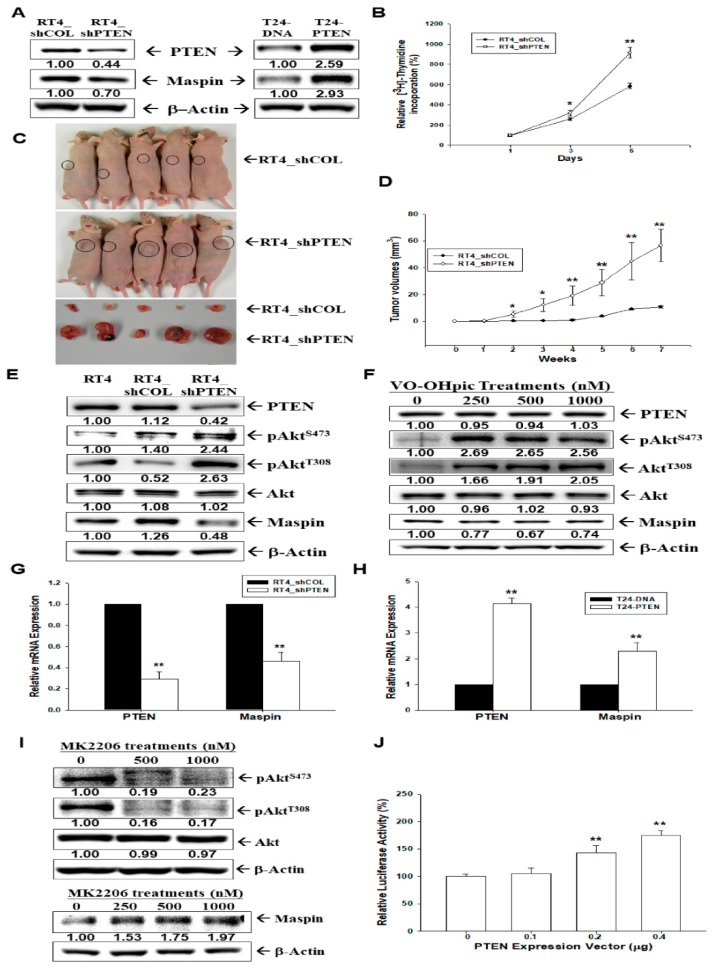
Modulation effect of PTEN on maspin expression in bladder carcinoma cells. (**A**) The mock-knockdown RT4 (RT4_shCOL), PTEN-knockdown RT4 (RT4_shPTEN), mock-transfected T24 (T24-DNA), and PTEN-overexpressed T24 (T24-PTEN) cells were lysed, and the protein levels of maspin and PTEN were determined through immunoblot assays. (**B**) The cell proliferation of RT4_shCOL and RT4_shPTEN was determined through the ^3^H-thymidine incorporation assays (±SE, *n* = 4). (**C**) Four-week-old male athymic nude mice were divided randomly into two groups. Cells (5 × 10^6^) were injected subcutaneously in the dorsal area of the mice (*n* = 6), and tumors derived from RT4_shCOL and RT4_shMapsin cells were measured after sacrifice. (**D**) The tumor volumes (±SE, *n* = 6) were measured every week during the indicated period. The expressions of PTEN, Akt, pAkt^S473^, pAkt^T308^, and maspin in RT4, RT4-shCOL, and RT4-shPTEN cells (**E**) as well as in VO-OHpic-treated RT4 cells (**F**) were determined through immunoblot assays. The mRNA ratios of PTEN and maspin between (**G**) RT4_shCOL and RT4_shPTEN cells and between (**H**) T24-DNA and T24-Maspin cells (±SE, *n* = 4) were determined by RT-qPCR assays. (**I**) The expressions of Akt, pAkt^S473^, pAkt^T308^, maspin, and β-Actin in MK2206-treated T24 cells were determined through immunoblot assays. (**J**) The reporter activity of maspin reporter vector cotransfected with various dosages of PTEN, as indicated in the HT1376 cells. Data are expressed as the mean percentage ± SE of luciferase activity relative to the mock-transfected group (*n* = 6). * *p* < 0.05, ** *p* < 0.01. The numbers indicate the ratio of the target gene/β-Actin, pAkt^S473^/Akt, or pAkt^T308^/Akt in relation to RT-4, or T24 cells.

**Figure 7 cancers-12-00010-f007:**
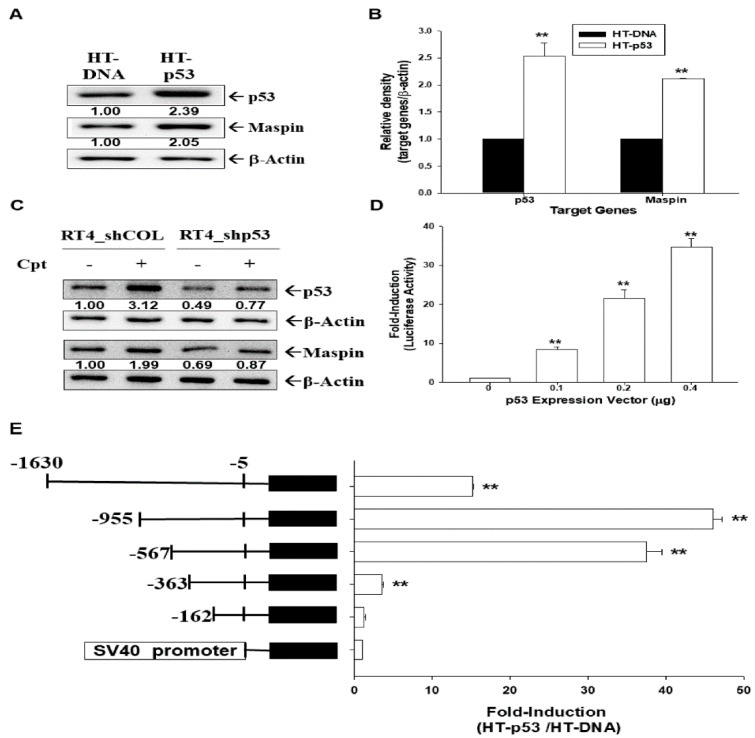
Modulation effect of p53 on maspin expression in bladder carcinoma cells. The mock-transfected HT1376 (HT-DNA) and p53-overexpressed HT1376 (HT-p53) cells were lysed, and the protein levels of maspin, p53, and β-actin were determined through (**A**) immunoblot and (**B**) RT-qPCR assays (±SE, *n* = 3). The numbers indicate the ratio of target gene/βActin in relation to HT-DNA cells. (**C**) The mock-knockdown RT-4 (RT4_shCOL) and p53-knockdown RT-4 (RT4_shp53) cells were treated with Cpt (1 μM) for 16 h. The cells were lysed, and the protein levels of maspin, p53, and β-actin were determined by immunoblotting. The numbers indicate the ratio of target gene/β-Actin in relation to RT4_shCOL cells. (**D**) The reporter activity of the maspin reporter vector was cotransfected with various dosages of p53, as indicated in the HT1376 cells. Data are expressed as the mean percentage ±SE (*n* = 6) of luciferase activity relative to the mock-transfected group. (**E**) Relative luciferase activity of HT1376 cells cotransfected with nested deletion constructs of the maspin reporter vector and the p53 expression vector. Data are expressed as the mean fold ± SE of luciferase activity induced by p53 relative to the mock-transfected group (±SE, *n* = 6). ** *p <* 0.01.

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
