# Peer review of "Maspin is a PTEN-Upregulated and p53-Upregulated Tumor Suppressor Gene and Acts as an HDAC1 Inhibitor in Human Bladder Cancer"

_cancers, 2019, doi:10.3390/cancers12010010_

Round 1

Reviewer 1 Report

The submitted manuscript by Lin and colleagues titled “Maspin is a PTEN- and p53-upregulated tumor suppressor gene and acts as an HDAC1 inhibitor in human bladder cancer” tries to provide data to clarify the role of maspin in regulating cellular proliferation.  There is a controversial history of reports for the role of maspin expression being correlated with increased tumor grade in clinical samples, but inhibitory in cellular growth assays.  The data provided indicate a potential role of PTEN and p53 in regulating maspin gene expression, but do not go far enough to draw definitive conclusions.  The data touch on many different aspects of maspin biology without providing enough data to support any one aspect.

There is a large body of literature already published examining the roles of PTEN and p53 in regulating maspin expression in bladder and various other solid tumors.  It is not clear how the data presented here move beyond those already published studies.  Much of the discussion states the data presented in this paper match previously published work.

Many of the conclusions drawn from the data provided are gross over interpretations lacking enough supporting evidence.  For example, a simple IF image such as that provided in Supplementary figure 1 cannot provide detailed localization information to conclude where maspin is localized in the bladder cancer cell lines.  A confocal image which accounts for the Z plane as well as a second cellular marker would be the minimum required for this.  Adding biochemical analysis of cellular fractions would support defining the cellular localization of maspin, but would still require more supporting experiments. Another example is the jump from maspin-regulated HDCA1 activity to gene expression levels is not supported.  There are no direct assessments of the link between maspin-regulated p21 expression, for example, and maspin-regulated HDAC1 activity.  As presented this data is a true, true, and unrelated logical outcome that is never directly tested.

The introduction would benefit from providing a clearer context of the stated controversy on maspin function in cellular transformation.  Several statements do not make sense and are most likely missing addition details needed to understand what is intended.  One example is in lines 53-54, it is not clear what is meant by lower maspin expression in the T24 cells. 

The paper would benefit from a model figure summarizing the described pathways so readers can understand what parts of maspin regulation and expression are being tested.

Overall the Results Section lacks context for the experiments being described making the data difficult to understand. As an example, given that the experiments switch between the cell lines, the Results section text needs to provide more detail on why a particular cell line is used in an experiment.  It becomes very confusing as to why a certain cell line is used and not another.  For example, why use RT4 cells in Figure 2A instead of the TSGH-8301 or HT1376 cells?

Figures 2 lacks controls demonstrating that the indicated knockdown or overexpression constructs alter maspin levels in the indicated cell lines.

It is not clear why two different EdU assays are shown in Figure 2A-D.  Both are validated assays for demonstrating EdU incorporation and the methods do not indicate that the two assays are testing different experimental questions.  The data in figure 2C and D requires a Maspin stain as a control.  This will allow accurate assessment of if the cells with maspin expression changes actually have a change in EdU incorporation.  This would provide a more accurate assessment of maspin expression levels on DNA synthesis in bladder cancer cells.

There is no explanation of why qRT-PCR for p21, cyclin D, vimentin is performed on the in vivo tumor tissues in Figure 4.

Section 2.7 gives no context to the experiments shown in figure 5.  Why is acetyl-H3 important in maspin biology?  The layout of the data is not logica,l and given the lack of explanation is extremely difficult to understand.

Other Points

The TMA in Figure 1C is too small to properly evaluate.  It might be better to provide larger representative images of the staining and provide a larger image of the TMA in a Supplemental Figure.

Both Supplemental Figures 1 and 2 need legends of to describe what is being shown.  Supplemental Figure 2 is not helpful as there are no labels on the lanes of the westerns.

There is a disconnect in the intensity scoring described in the methods, 0, 1, 2, 3 and the graph in figure 1D which shows normal tissue with an intensity of almost 4.  Also, the method of scoring the TMA is vague.  Section 4.3 seems to indicate that a software program PAX-it was used, but this is not an automated scoring software package.  More detail needs to be provided.

The MTS assay needs a reference or at least the identity of the manufacturer so the protocol can be looked up by the reader. 

The comparisons of the HDAC1 activity data in Figure 5B and C is confusing to interpret as presented.  It would be better to graph the fold change and provide the IgG and HDAC1 IP in a supplemental table or graph.

The western blots in Figures 6 are not convincing in demonstrating that changes in PTEN levels lead to a change in maspin levels.  Densitometrical analysis would help demonstrate changes at the protein level as was done in figure 7 A and B.

The Methods used for the Luciferase reporter assay are not provided in the Methods section.  What version of the luciferase-driven maspin promoter plasmid was used in Figures 6 J and 7D?  It is not clearly stated in the Results, Figure Legends or Methods sections.

The statistical analysis section states ANOVA analysis is used, but does not list a post hoc analysis to correct for multiple comparisons.  This is required.

Reviewer 2 Report

In this manuscript the authors report that Maspin, a member of the the B-serine protease inhibitor superfamily, acts as tumor suppressor in bladder carcinogenesis.

They demonstrate that Maspin is upregulated by PTEN and p53, whereas is downregulated by AKT.

 Finally, they report that Maspin inhibits HDAC1. Indeed, Maspin silencing downregulates the levels of acetyl-histone H3.  

The study is well planned.

Even though the suppressor role of Maspin has been already reported in several human neoplasias, the authors unveil some mechanisms concerning the its regulation iby PTEN and P53, and  iinhbition of HDAC1.

Criticism

1- All the Figures are of poor quality and need  to be reformatted as regards the relative size

2- All the results are written without the proper introduction of the scientific rationale and seem more like a list of notions.

3- In Figure 1A and B there is an apparent discrepancy between what is written in the text and what is shown in the figure (box plot). It is also necessary to describe immunohistochemical staining in the results and indicate which the normal controls are.

4- Please also show in Figure 2 the Maspin silencing and over-expression by western blot.

5- In figure 5A please explain why beta-actin is shown three times.

6- The Materials and Methods section is not very accurate. Some methods should be extended, while the others could be moved to Supplementary Materials.

Reviewer 3 Report

In this original article the authors aimed to evaluate the role and expression of maspin in bladder cancer (both in vivo and in vitro). The idea is interesting and the study well conducted. However, some clinic-pathologic variables and info are missing and the methods and results should be reported more clearly. Some suggestions to improve the quality of the manuscript are reported:

Major

Introduction

I do not agree with the correlation between the lack of effective strategies for early detection and the high recurrence and mortality rates of bladder cancer. Actually, recurrence rates are mainly inherent the biology of the cancer and the lack of effective intravesical therapies (the most effective intravesical therapy is BCG and is the same agent since almost 50 years). Therefore, understanding the molecular mechanism of new suppressor genes is essential maybe not for the diagnostic setting but to pave the way for new selective drugs and to identify biomarkers able to predict the response to therapy. The second part of the introduction (from row 43) reporting differing results from previous studies is quite confusing and should be revised.

Methods

It would be useful to know if the bladder tumor samples derive from endoscopic resections (TURBs) or radical cystectomy specimens. How have been the normal bladder tissues collected? From normal mucosa from patients with bladder cancer? Or from healthy controls? This should be stated in the methods section

Results

It would be interesting to know if maspin expression correlates also with tumor stage and not only with tumor grade. How many tissue samples were collected? How many analyzed? It would be useful to build a table to report the clinico-pathologic characteristics of the patients related to the samples.

Minor

Abstract:

Row 20: “…than the paired normal tissues.”. The term “in” is missing. Row 25: please specify akt (i.e. protein kinase B).

Reviewer 4 Report

The paper by Lin and colleagues is informative and interesting. It proposes maspin as a tumor suppressor gene in bladder cancer. Authors have performed complex network of experiments and better describing the sequence could help in the reading. Indeed, this is a well designed study, but a schematic figure of design could help in following the workflow of the paper.

One minor observation is the poor resolution of figures, especially figure 2.

Round 2

Reviewer 1 Report

The authors have addressed most of the concerns raised in the original review.  However, some concerns raised including a couple major issues have not been addressed.  These issues cannot be resolved by altering the text of the manuscript- 

1 - There are still large gaps in the logic of the argument presented making the manuscript difficult to understand.  The data touch on many different aspects of maspin biology without providing enough data to support any one aspect.

2 - The cellular localization of maspin is missing critical controls.  The data as presented (Sup Fig 2) does not support any conclusions on maspin localization in the cell.  It is also not clearly explained in the manuscript why understanding the subcellular localization of maspin is necessary.

3 - There is no additional data provided to confirm or disprove the link between maspin-regulated HDAC activity and maspin-regulated gene levels.  These are loosely correlated results that do not support the conclusions stated in the paper.

4 - It is still not clear why the 2 EdU assays in Figure 2 were performed.  What is the mechanistic difference between the two assays?  Why is presenting data from both assays necessary?  What does one assay say about maspin's effects on DNA synthesis that the other does not?

5- The added language to the statistical methods section 4.18 "The post-hoc analysis was used to correct for multiple comparisons" is meaningless.  There are numerous types of post hoc analysis that can be done to an ANOVA.

Reviewer 3 Report

I would like to thank the authors for their reply and for having improved the quality of the manuscript
